# LHAT-YOLO: Study on intelligent monitoring algorithm for helmets at construction sites

**Jun Wang** [1]*, **Dongsheng Zhao**[1], **Haoran Jiang**[2], **Peng Lin**[3], **Xin Tang**[4]

1 Sichuan Institute of Industrial Technology, Deyang, Sichuan, China, 2 Chongqing Juneng Construction Group Co., Ltd, Chongqing, China, 3 The Housing Expropriation Center of Dazhou Eastern Economic Development Area Management Committee, Dazhou, Sichuan, China, 4 School of Civil Engineering, Chongqing Three Gorges University, Chongqing, China

* junwang20252025@163.com

## Abstract

Construction sites in civil engineering projects are prone to sudden accidents. In particular, head injuries pose a significant safety threat to construction workers. Helmets play a vital role in protecting the heads of construction workers. Most construction sites still rely on manual methods to monitor workers' helmet compliance, which is not only inefficient but also incapable of real-time monitoring. While traditional models can achieve intelligent monitoring, their lack of real-time capability fails to meet practical demands. However, the introduction of deep learning has transformed this situation. Therefore, this study proposes an intelligent monitoring method for helmet wearing at civil engineering construction sites based on deep learning theory. The study uses GSConv to improve the convolution module of the YOLOv11 deep learning model and adds a lightweight detection head: FCD (Fast Convolutional Detection) to establish the LHAT-YOLO model (Lightweight YOLO model for detecting helmets). While reducing the complexity of the model, it maintains accuracy and achieves efficient and intelligent detection of helmet wearing at construction sites. Experimental results show that on the dataset comprising 19,780 images in the training set, 2,473 images in the validation set, and 2,473 images in the test set, the LHAT-YOLO model reduces GFLOPs by 11% and Params by 9.5% compared to the YOLOv11 model. Overall, the LHAT-YOLO model achieves a Precision value of 93.95%, a Recall value of 88.99%, an mAP50 value of 94.92%, and an mAP50-95 value of 65.28%. This demonstrates that the LHAT-YOLO model maintains high accuracy even while being lightweight.

## 1 Introduction

With the growing demand in the construction industry, safety management at civil engineering construction sites has become increasingly important [1–4]. The correct use of helmets can effectively protect the safety of construction workers. It is

**Data availability statement:** All relevant data are within the paper and its Supporting Information files.

**Funding:** The author(s) received no specific funding for this work.

**Competing interests:** The authors declare that they have no known competing financial interests or personal relationships, which will not affect the work reported in this article.

especially necessary to wear helmets at construction sites where accidents are frequent. However, a report by the US National Safety Council shows that there were more than 65,000 head injuries and 1,020 deaths at construction sites due to not wearing helmets [5]. Data from the Brain Injury Association shows that although head injuries account for more than 20% of all injuries, only 3% of personal protective equipment purchased is used for head protection [6]. Therefore, it is very necessary to supervise and manage whether construction workers wear helmets [7–9]. Many countries have included safety issues at civil engineering construction sites on their list of key concerns. At the same time, they have invested a lot of manpower, material resources, and financial resources in related research and practice, with the aim of effectively reducing the incidence of safety accidents [10–12].

Traditional management methods mainly rely on human supervision, that is, arranging personnel to patrol and supervise different areas. This method is not only inefficient and unable to achieve comprehensive supervision, but also consumes a lot of time and human resources. With the continuous development of deep learning object monitoring technology, deep learning-based helmet detection technology has emerged. This not only has significant advantages in terms of improving efficiency, but also greatly saves human resources [13–16]. Currently, many scholars are conducting in-depth research on intelligent helmet recognition, including motorcycle helmets, bicycle helmets, etc [17–20]. Different scholars have proposed different research methods for intelligent helmet detection based on actual usage scenarios at construction sites such as coal mines and civil engineering projects [21–26]. Two-stage object detection models are mostly used in scenarios that require high accuracy, but they cannot meet the real-time detection requirements of construction sites [27]. However, one-stage object detection models are known for their speed, especially the YOLO series and SSD, which are widely used in intelligent helmet detection projects [28–31]. Therefore, how to further reduce the number of model parameters and the amount of computation while maintaining the high accuracy of the model is currently a hot topic of research. Under normal circumstances, after a model is lightweighted, its accuracy will decrease. There are three main reasons for this. First, some objects are located in complex environments, and it is difficult for lightweight models to accurately extract the features of such complex objects. Second, some objects are small in size, and current object detection models mainly focus on large and medium-sized objects and are not specifically optimized for small objects. Third, the GFLOPs and Params of the model will be reduced after lightweighting, which will cause errors in the process of extracting object features.Safety helmet monitoring at civil engineering construction sites has high requirements for the real-time performance and accuracy of deep learning models. Therefore, it is practical to achieve efficient and accurate safety helmet wearing detection while lightening the model.

In summary, this study addresses the critical need for enhancing the systematic real-time performance of hard hat detection at civil engineering construction sites by proposing a lightweight, high-precision LHAT-YOLO model. Building upon the top-performing YOLOv11 architecture, this model deeply integrates two key innovations. First, it introduces GSConv modules to replace portions of standard

convolutions, significantly reducing computational complexity and parameter count. Second, the FCD detection head is introduced, effectively improving the localization and recognition accuracy of hard hats in complex backgrounds. Through exhaustive experiments, this study validates the model's superior performance on a self-built hard hat dataset, providing a robust theoretical foundation for intelligent safety management in civil engineering construction and other high-risk industries.

## 2 Methods and data

### 2.1 Data source

The study collected images of construction sites through construction site cameras and cameras, and combined them with images collected from the Internet to form a high-quality dataset of 24,726 images. The Labelimg tool was used to label the images (Fig 1). The label types were divided into two categories, "helmet" and "person," to monitor whether construction workers on site were wearing helmets correctly. The dataset was randomly split into a training set, validation set, and test set in an 8:1:1 ratio. The training set contains 19,780 images, the validation set contains 2,473 images, and the test set contains 2,473 images.

### 2.2 Evaluation criteria

To validate the performance of the improved model, the following metrics were used: recall (R), precision (P), mean average precision (mAP), number of parameters (Params), and computational complexity (GFLOPs). Params refers to the total number of training parameters in a neural network model. GFLOPs denotes the number of floating point operations per forward pass at the specified input size; we report theoretical complexity estimated from the architecture. The recall

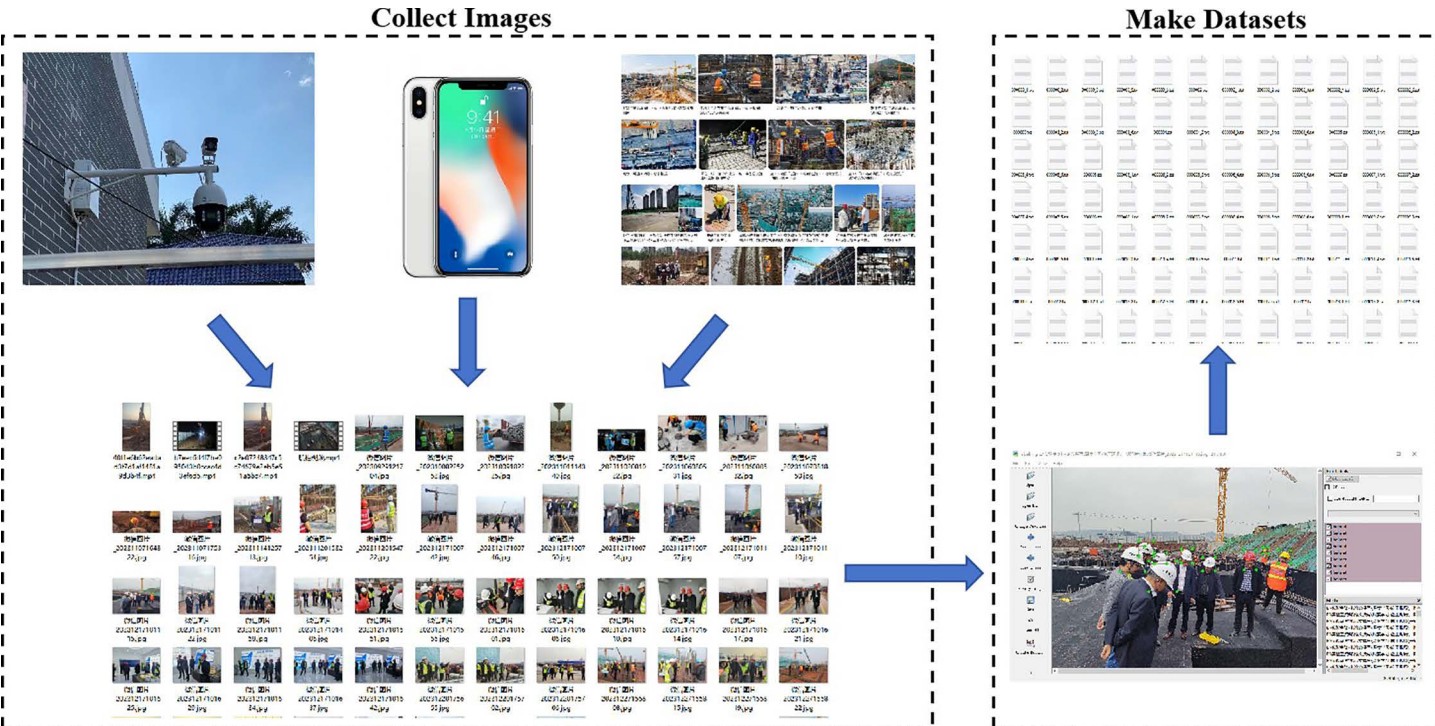

**Fig 1. Data set preparation flowchart.**

indicator is used to evaluate the model's prediction effect on actual helmet-wearing samples in the helmet-wearing detection task. It represents the proportion of samples correctly identified by the model as wearing helmets out of all actual helmet-wearing samples. The calculation formula is as follows:

$$Recall = \frac{TP}{TP + FN} \tag{1}$$

The accuracy rate indicator is used to evaluate the accuracy of the model in predicting whether a person is wearing a helmet in a helmet wearing detection task. It represents the proportion of samples correctly predicted by the model as wearing a helmet out of the total number of samples predicted as wearing a helmet. The calculation formula is as follows:

$$Precision = \frac{TP}{TP + FP} \tag{2}$$

Among them, TP refers to the number of positive samples detected as true. FP refers to the number of negative samples predicted as true. FN refers to the number of positive samples not detected.

The mAP metric is divided into two forms: mAP50 and mAP50-95. mAP50 represents the average precision when the IoU threshold is fixed at 0.5. mAP50-95, on the other hand, is the average precision calculated across multiple IoU thresholds (from 0.5 to 0.95, with intervals of 0.05). IoU is a metric that measures the degree of overlap between predicted boxes and ground truth boxes. The formula is:

$$AP = \int_0^1 P(R)dR \tag{3}$$

$$mAP = \frac{1}{N}\sum_{i=1}^{N} AP \tag{4}$$

## 2.3 Experimental environment and parameter settings

To ensure consistency in the environment variables during model training, this study conducted all model training in the same environment. The experimental environment is shown in Table 1. The resolution of the input images is 640×640. The initial learning rate is 0.01. The decay coefficient is 0.0005.

**Table 1. Experimental environment and parameters.**

| Name | Configuration | Name | Configuration |
|---|---|---|---|
| System | Windows 10 | CUDA | 11.6 |
| GPU | NVIDIA GeForce RTX 4070 Ti | Batch_size | 24 |
| CPU | 13th Gen Intel(R) Core(TM) i9-13900K | Image size | 640×640 |
| Video memory | 12G | Learning rate | 0.01 |
| RAM | 128G | Decay coefficient | 0.0005 |
| Development environment | Python3.9 | Num classes | 2 |
| Development framework | PyTorch1.18 | Epoch | 100 |

## 3 Model Improvement and Innovation

### 3.1 YOLOv11 model

The YOLO series of algorithms is currently one of the fastest growing and most effective object detection algorithms. YOLOv11 is one of a new generation of object detection algorithms developed by Ultralytics [32]. It features significant improvements in architecture and training methods over previous versions of YOLO. It integrates an improved model structure design, enhanced feature extraction technology, and optimized training methods (Fig 2). The C3k2 module is a computationally more efficient implementation of the Cross Stage Partial (CSP) Bottleneck. The "k2" in C3k2 denotes a smaller convolution kernel size [33]. YOLO11 retains the SPPF module but introduces a new Cross Stage Partial with Spatial Attention (C2PSA) module following it [34]. What truly sets YOLO11 apart is its impressive combination of speed and accuracy, making it one of the most powerful models Ultralytics has created to date. Through improved design, YOLO11 offers better feature extraction, enabling it to capture complex aspects more accurately even in challenging scenarios.

### 3.2 GSConv and FCD modules

GSConv (Grouped and Shuffled Convolution) is an innovative convolution method designed for lightweight applications [35]. By combining the advantages of standard convolution (SC) and deep separable convolution (DSC), GSConv achieves an optimized balance between computational efficiency and feature representation capability (Fig 3). Its core architecture employs a grouped shuffled convolution structure: the input feature map is divided into two parts, with standard convolution preserving channel information and deep separable convolution reducing computational complexity. Channel shuffling is then used to enable cross-group feature interaction, effectively mitigating the feature fragmentation issues inherent in DSC. This technique reduces computational complexity while maintaining accuracy close to that of the

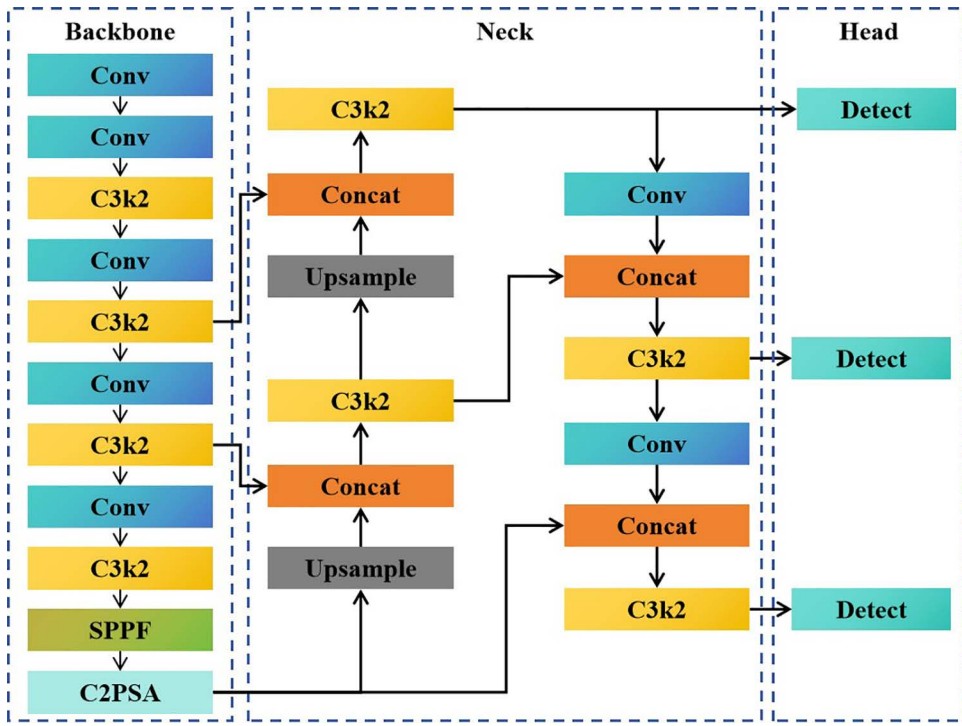

**Fig 2. Network structure diagram of the YOLOv11 model.**

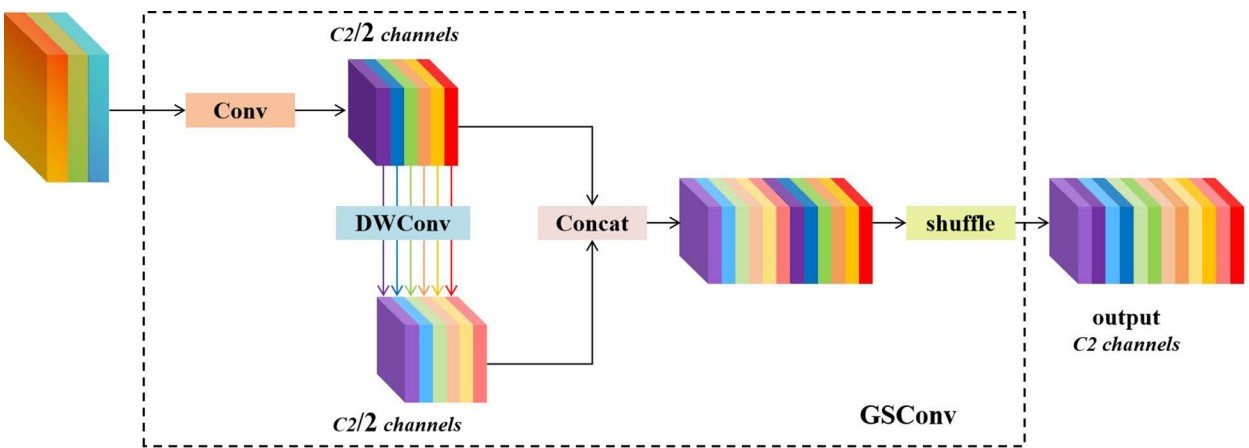

**Fig 3. Structural diagram of the GSConv module.**

original model. GSConv is particularly well-suited for mobile and edge computing scenarios. It offers a new approach to lightweight model design that balances efficiency and accuracy.

In the original YOLOv11 detection head, each detection layer performs convolutional operations independently, which leads to an increase in the overall number of parameters and computational complexity of the model. This has a significant impact on the practical application of the model in resource-constrained environments such as embedded devices. This paper proposes a lightweight shared convolutional detection head, FCD (Fast Convolutional Detection), to address this issue. By combining shared convolutions and detail-enhancing convolutions, the model's detection capabilities for small objects and complex backgrounds are effectively improved while significantly reducing the number of parameters and computational complexity (Fig 4. a). The core of the FCD design lies in the synergy between the shared convolution module and the detail-enhanced convolution module. It can achieve efficient use of computing resources and meet the needs of real-time detection of helmets at construction sites. The FCD module is a lightweight shared detection head designed to reduce parameter redundancy across multi-scale branches. Input multi-scale feature maps (C3, C4, C5) from the backbone network have channel counts of 128, 256, and 512 respectively. Each branch first passes through a 3 × 3 convolution + GroupNorm + SiLU activation (Conv_GN) to unify channels to 256 dimensions. High-level features are then extracted via a shared convolution block, comprising a depthwise separable 3 × 3 convolution (number of groups = number of channels) and a 1 × 1 convolution, both followed by GroupNorm and SiLU activation. The features are then fed into the bounding box regression branch and the classification branch. The bounding box branch uses a 1 × 1 convolution to output 4 × Reg_max = 64 channels (Reg_max = 16), achieving high-precision bounding box regression via a DFL layer. The classification branch uses a 1 × 1 convolution to output category predictions. Finally, the prediction results from different scales are spatially concatenated and decoded using anchor decoding to obtain the final detection results.

In addition, FCD introduces a scale layer in each detection layer. The scale layer adjusts the feature amplitude and offset through dynamic linear transformation to ensure that the model can meet the detection requirements of objects of different scales. Lightweight design of the model may reduce its ability to extract features, thereby affecting the detection accuracy of the model. Therefore, this paper adds a DeConv module (Detail-Enhanced Convolution) to FCD. The DeConv module combines standard convolution with multiple difference convolutions to capture local gradient information of the input features, thereby enhancing the model's ability to perceive detailed features. The DeConv module consists of five parallel paths. These include standard convolution Conv2d and four types of Difference Convolution. Difference Convolution includes Central Difference Convolution (CDC), Horizontal Difference Convolution (HDC), Vertical Difference Convolution (VDC), and Angular Difference Convolution (ADC) (Fig 4. b).

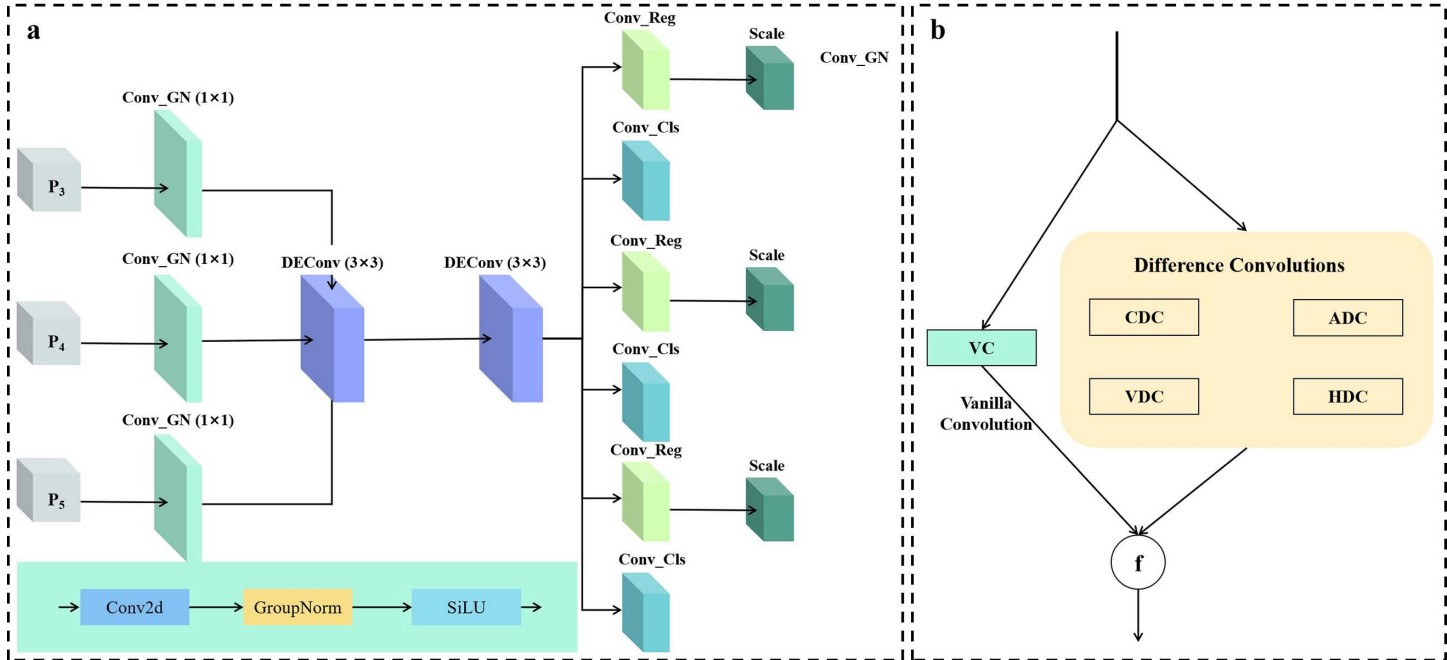

**Fig 4. Structural diagram of the FCD detection head module.**

## 3.3 LHAT-YOLO model

Due to the complex and changing environment at the construction site, the study chose to add a complex object detection layer to the YOLOv11 model to improve the model's detection accuracy and ability to handle complex environments. The complex object detection layer introduces an additional detection scale P2 (stride 4) within the YOLOv11 detection head. The P2 features are obtained by upsampling (×2) the P3 features, then concatenated and fused with shallow features from the backbone network (stage 2 output, stride＝4). This combined input undergoes feature refinement through a C3k2 module. Ultimately, the detection head expands from three scales (P3, P4, P5) to four scales (P2, P3, P4, P5), thereby enhancing detection performance for small-scale complex objects.

However, this would greatly increase the overall number of parameters and computational load of the model. Therefore, the study chose to replace part of the Conv in the backbone with GSConv. This method reduces the number of parameters and the computational load of the model backbone to achieve the goal of overall model lightweighting. At the same time, the FCD module is used to replace the Detect module of the YOLOv11 model. This not only effectively integrates feature information from different levels and maintains the high accuracy of the model, but also further reduces the number of parameters in the model and improves its detection efficiency. The design of the FCD module also reduces information loss, improves feature transmission efficiency, and helps optimize model performance. The structure of the LHAT-YOLO object detection model is shown in Fig 5.

## 4 Results and discussion

### 4.1 Ablation experiments

An ablation experiment was conducted on the helmet dataset to verify the performance of the LHAT-YOLO model in helmet detection (Table 2). Each set of experiments was conducted under the same experimental parameters and environment. The experiments in Group A showed that the YOLOv11n model performed excellently in helmet

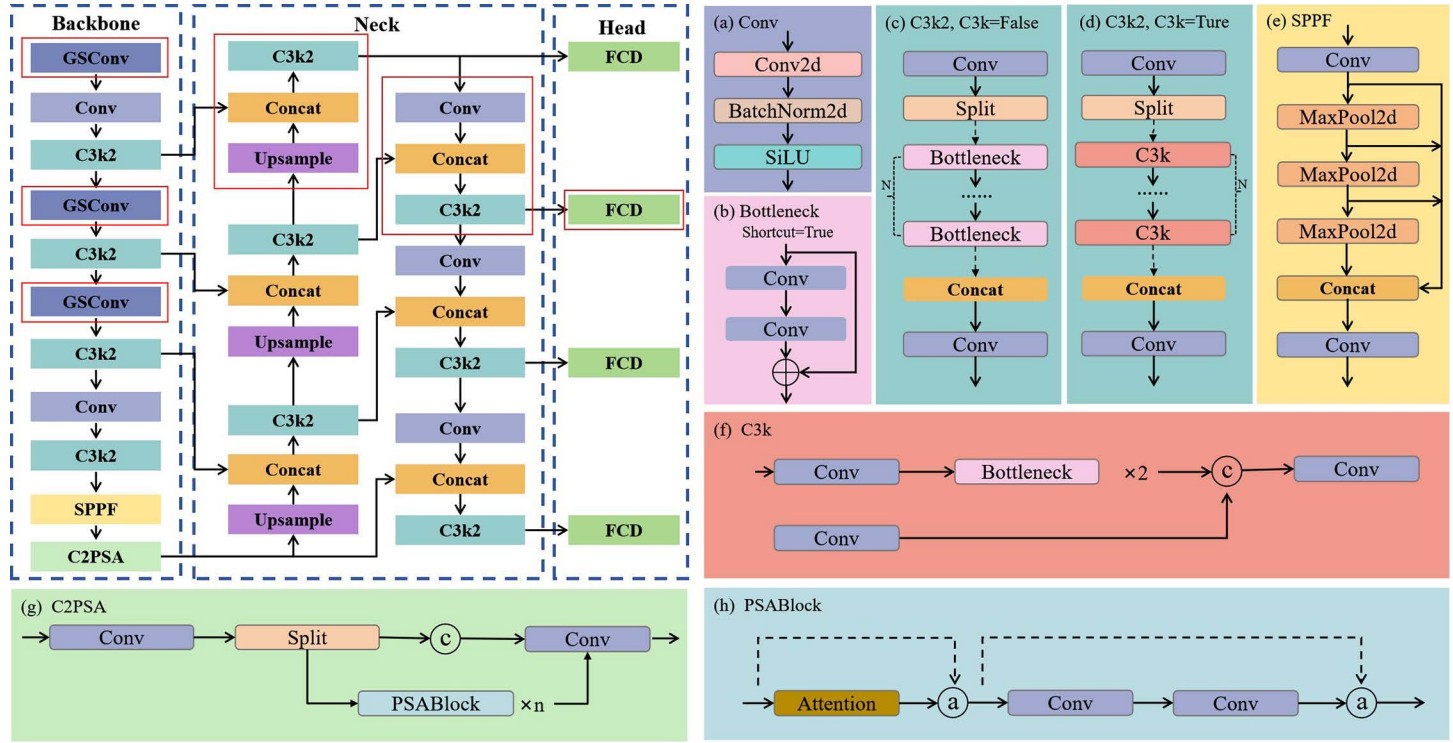

**Fig 5. Network structure diagram of the LHAT-YOLO model.**

**Table 2. Ablation experiment results.**

|   | YOLOv11n | Detection layer for complex objects | GSConv | FCD | Precision/% | Recall/% | mAP50/% | mAP50-95/% | Params/M | GFLOPs |
|---|---|---|---|---|---|---|---|---|---|---|
| A | √ | × | × | × | 93.62 | 87.56 | 94.21 | 64.73 | 2.59×10⁶ | 6.4 |
| B | √ | √ | × | × | 93.31 | 87.99 | 94.80 | 65.21 | 2.67×10⁶ | 10.4 |
| C | √ | √ | √ | × | 93.58 | 87.62 | 94.50 | 64.46 | 2.51×10⁶ | 10.1 |
| D | √ | √ | × | √ | 93.34 | **89.3** | **95.19** | **65.67** | 2.51×10⁶ | 6 |
| E | √ | √ | √ | √ | **93.95** | 88.99 | 94.92 | 65.28 | **2.34×10⁶** | **5.7** |

detection. The experiments in Group B proved that the accuracy of the model was improved after adding the Detection layer for complex objects. However, after adding the Detection layer for complex objects, the GFLOPs and Params of the model increased significantly, which did not meet the requirements for real-time monitoring. Experiments in Groups C, D, and E demonstrate that the addition of the GSConv module and FCD module not only reduces the GFLOPs and parameters of the YOLOv11n model but also improves the model's detection accuracy. Specifically, GFLOPs are reduced by 11% compared to the YOLOv11n model, and parameters are reduced by 9.5% compared to the YOLOv11n model. The Precision index increased by 0.23, the Recall index increased by 1.03, the mAP50 index increased by 0.51, and the mAP50-95 index increased by 0.35. This proves that the LHAT-YOLO model can simultaneously achieve high performance and light weight, which is more suitable for helmet detection projects at actual construction sites.

## 4.2  Comparative experiments

To further validate the practicality and reliability of the LHAT-YOLO model, The training results for different models are shown in Fig 6 and the study selected multiple different models for training comparison (Table 3). As shown, the GFLOPs and Params of other models are both greater than the LHAT-YOLO model. Among them, the YOLOv9t model has the highest GFLOPs, the YOLOv8n model has the highest Params, and the LHAT-YOLO model has both lower GFLOPs and Params than other models. More importantly, these models performed worse than the LHAT-YOLO model on the helmet detection dataset. It can be seen that the LHAT-YOLO model is not only lighter, but also comparable to other models in terms of accuracy.

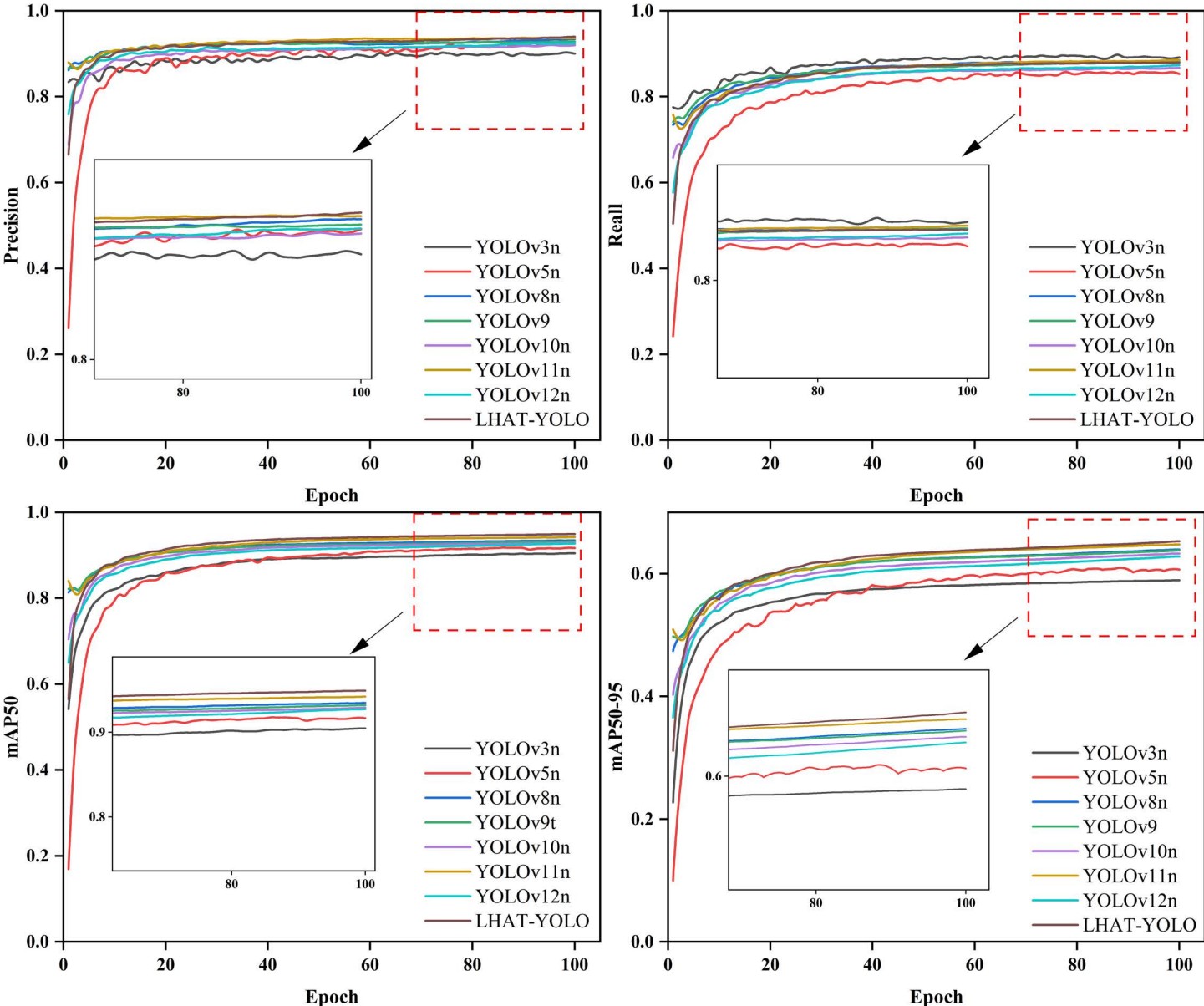

**Fig 6.  Network structure diagram of the LHAT-YOLO model.**

**Table 3. Comparison results of different models.**

| Model | Precision/% | recall/% | mAP50/% | mAP50-95/% | Params/M | GFLOPs |
|---|---|---|---|---|---|---|
| YOLOv3n | 90.56 | **89.79** | 90.47 | 58.92 | $7.64 \times 10^6$ | 12.4 |
| YOLOv5n | 91.24 | 85.35 | 91.75 | 60.64 | $9.10 \times 10^6$ | 21.6 |
| YOLOv8n | 93.35 | 87.03 | 93.47 | 63.92 | $3.01 \times 10^6$ | 8.2 |
| YOLOv9t | 92.81 | 87.13 | 93.19 | 63.77 | $2.66 \times 10^6$ | 11 |
| YOLOv10n | 91.96 | 85.70 | 92.89 | 63.26 | $2.71 \times 10^6$ | 8.4 |
| YOLOv11n | 93.62 | 87.56 | 94.21 | 64.73 | $2.59 \times 10^6$ | 6.4 |
| YOLOv12n | 92.43 | 86.32 | 92.70 | 62.81 | $2.52 \times 10^6$ | 6 |
| LHAT-YOLO | **93.95** | 88.99 | **94.92** | **65.28** | **$2.34 \times 10^6$** | **5.7** |

## 4.3 Visual analysis

In order to further verify the practical application effect of the LHAT-YOLO model in civil engineering construction sites, the YOLOv11 model and the LHAT-YOLO model were used to detect helmets in images of different construction scenes. In night scenes, the LHAT-YOLO model performs better than the YOLOv11 model (Fig 7a, d). Not only can it detect smaller objects (Fig 6b, c), but it can also accurately detect helmets in night scenes with strong interference (Fig 7e, f). In scenes with obscuration, the LHAT-YOLO model can accurately detect objects with severe obscuration, while YOLOv11 misses some objects (Fig 7g, h, i, j, k, and l). However, nighttime construction and obstruction are particularly common phenomena at civil engineering construction sites, proving that the LHAT-YOLO model is more suitable for intelligent helmet detection projects at civil engineering construction sites. Similarly, the heatmap results also show that the LHAT-YOLO model can focus more accurately on the object and is less affected by interference factors in the environment (Fig 8).

To further validate the performance of the LHAT-YOLO model in complex scenes, the researchers selected images with more complex environmental content to test the model. The results showed that the LHAT-YOLO model also performed better than the YOLOv11 model (Fig 9). The YOLOv11 model not only missed detections but also made false detections in complex environments. The YOLOv11 model incorrectly detects the shadow on the ground as a "person" in Fig 8e and incorrectly identifies part of the excavator as a "person" in Fig 8h. Combined with the heatmap results, it can be seen that during the detection process, the YOLOv11 model's attention was distracted by complex environmental factors in the image, while the LHAT-YOLO model was still able to focus most of its attention on the object to be detected (Fig 10). This proves that the LHAT-YOLO model is better suited to the complex environment of civil engineering construction sites and can efficiently and accurately detect workers wearing helmets and those not wearing helmets. However, the LHAT-YOLO model still experiences a small number of missed detections and duplicate detections (Fig 7f, l and Fig 8l), indicating that there is still room for improvement in the model.

## 5 Conclusion

(1) This study proposes a model suitable for intelligent detection of safety helmets at construction sites: the LHAT-YOLO model. The LHAT-YOLO model is built upon the YOLOv11 architecture by incorporating a complex object detection layer, GSConv, and FCD head detection. It demonstrates enhanced resistance to interference in complex environments while maintaining a smaller overall parameter count and computational load. This achieves model lightweighting without compromising high-precision performance.

(2) The LHAT-YOLO model outperforms all other models across all metrics while maintaining the lowest number of parameters and computational complexity. Ablation experiments demonstrate that compared to the YOLOv11 model, LHAT-YOLO achieves an 11% reduction in GFLOPs and a 9.5% reduction in parameters. Precision, Recall, mAP50, and mAP50-95 improved by 0.33, 1.43, 0.71, and 0.55, respectively. Comparative experiments demonstrate that while

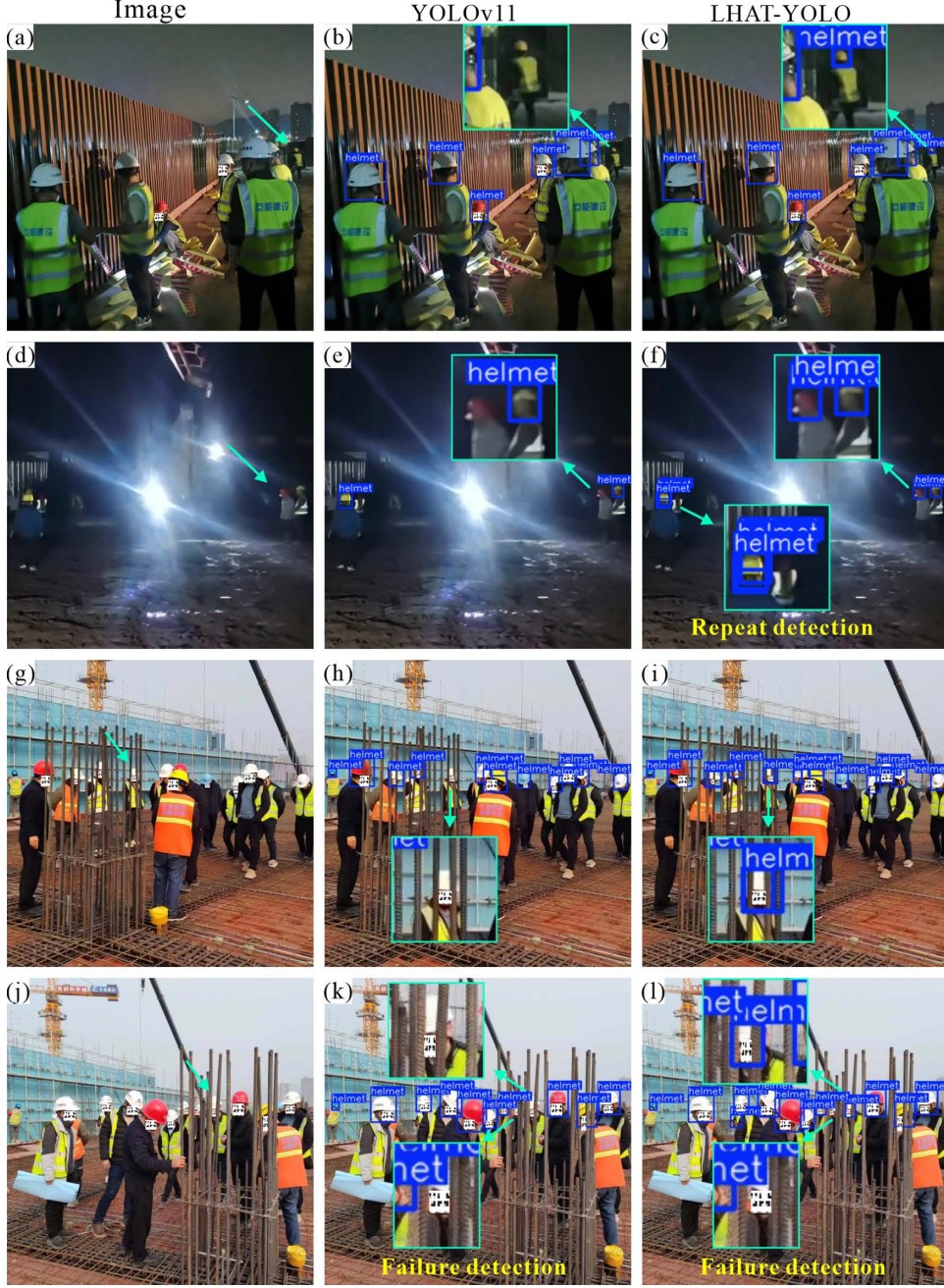

**Fig 7. Detection results for nighttime scenes and scenes with obstructions.**

the LHAT-YOLO model has the lowest number of parameters and computational requirements, it achieves the highest accuracy. This proves that the LHAT-YOLO model maintains high precision despite its lightweight design, making it fully capable of performing intelligent safety helmet detection at civil engineering construction sites.

(3) The LHAT-YOLO model can detect smaller objects in nighttime scenes and accurately identify safety helmets even in highly distracting nighttime environments. In occluded scenes, the LHAT-YOLO model accurately detects heavily

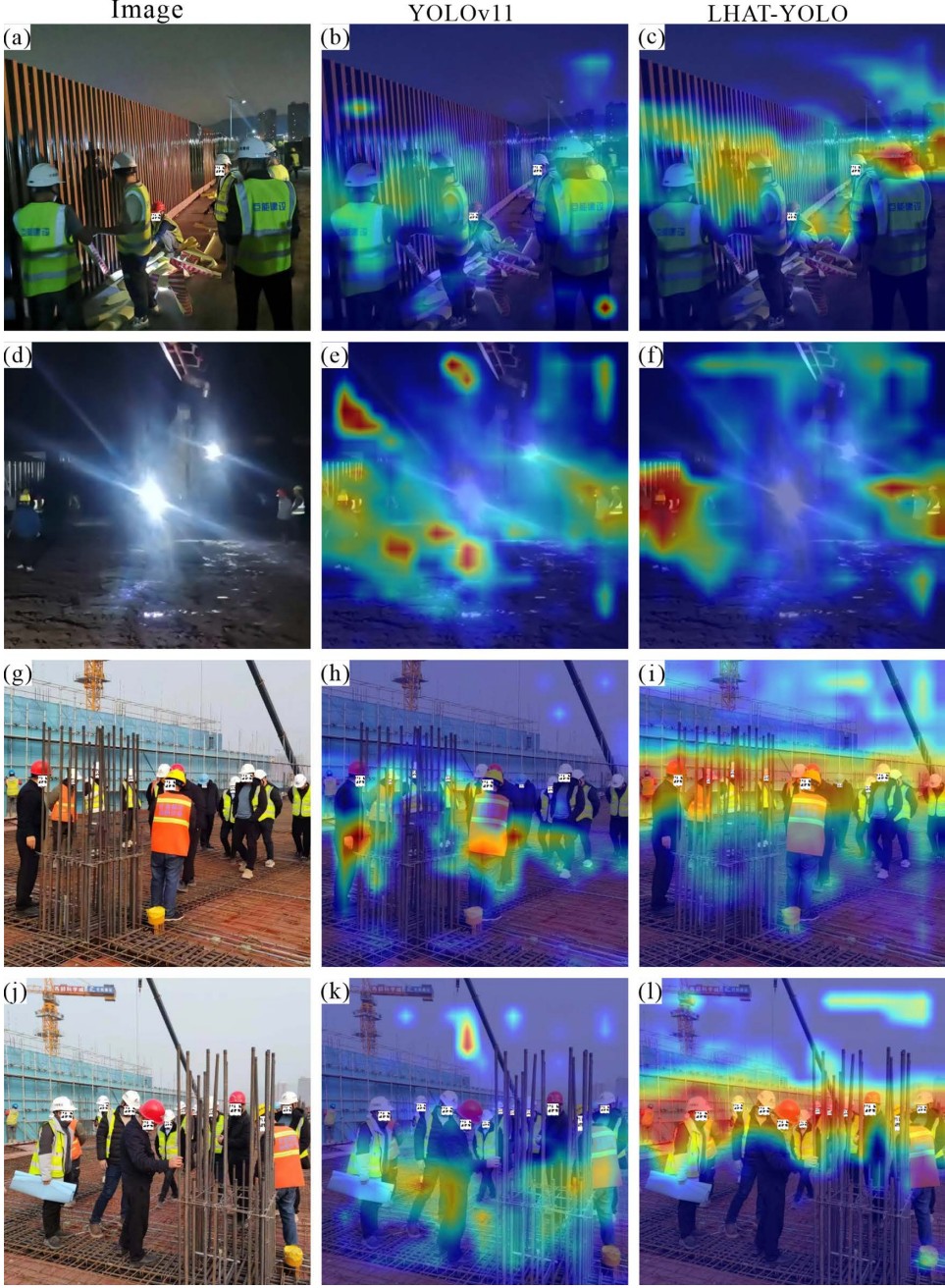

**Fig 8. Heatmap analysis of night scenes and scenes with obstructions.**

obscured targets, whereas YOLOv11 exhibits missed detections. Heatmap results indicate that when detecting hard hats in complex environments, the YOLOv11 model is significantly affected by environmental factors, whereas the LHAT-YOLO model can focus substantial attention on the target to be detected. This demonstrates that the LHAT-YOLO model better adapts to the complex environments of civil engineering construction sites, thereby enabling intelligent hard hat detection.

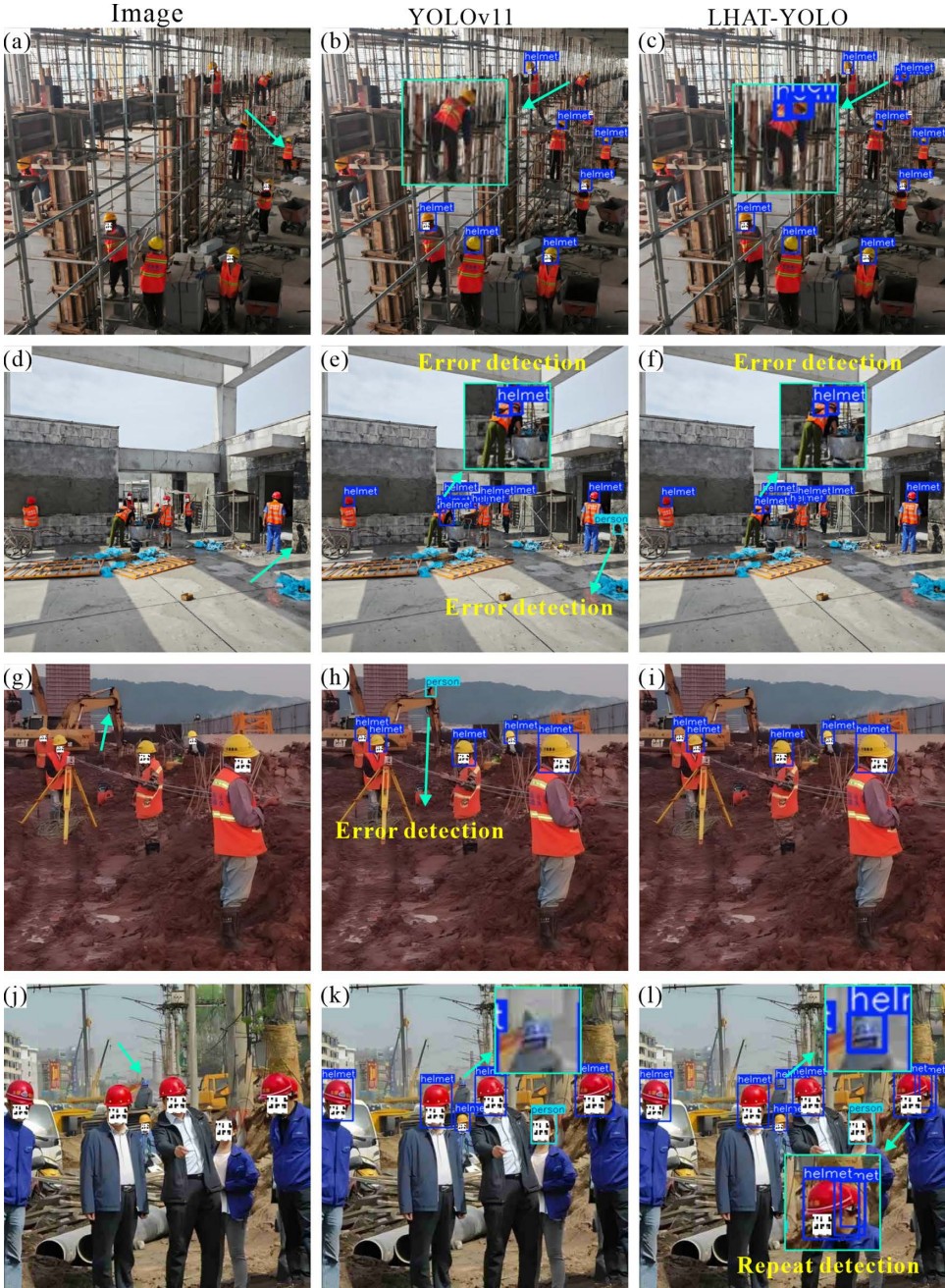

**Fig 9. Small object and complex scene detection results.**

## Supporting information

**S1 Data. Minimal data set.**

(ZIP)

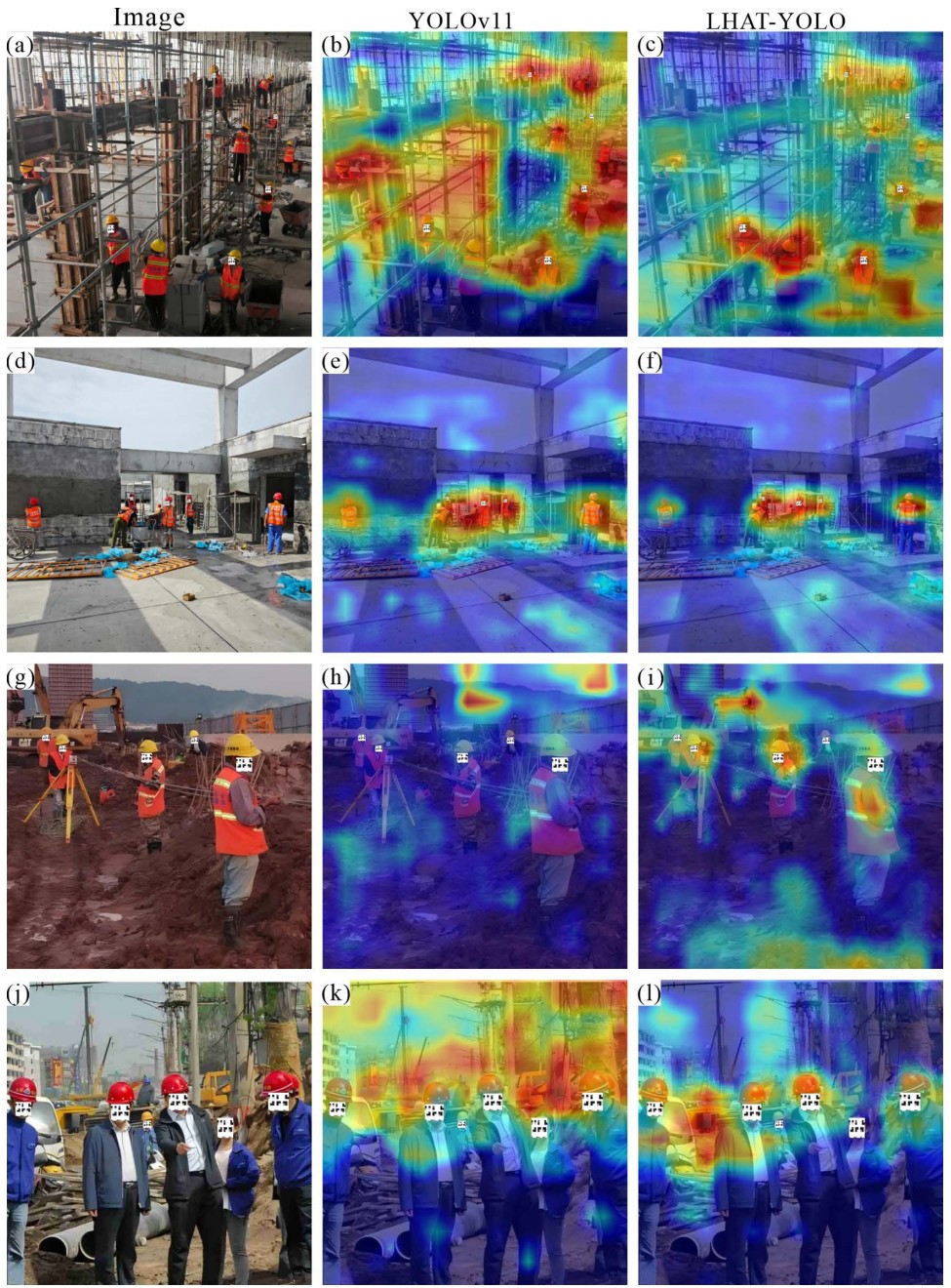

**Fig 10. Heatmap analysis of small objects and complex scenes.**

## Author contributions

**Conceptualization:** Jun Wang.

**Data curation:** Jun Wang.

**Formal analysis:** Jun Wang, Dongsheng Zhao.

**Funding acquisition:** Dongsheng Zhao, Haoran Jiang, Peng Lin, Xin Tang.

**Investigation:** Jun Wang.

**Methodology:** Jun Wang.

**Project administration:** Dongsheng Zhao, Haoran Jiang, Peng Lin, Xin Tang.

**Resources:** Jun Wang.

**Software:** Jun Wang.

**Supervision:** Jun Wang, Dongsheng Zhao, Haoran Jiang.

**Validation:** Jun Wang.

**Visualization:** Jun Wang.

**Writing – original draft:** Jun Wang.

**Writing – review & editing:** Jun Wang, Dongsheng Zhao, Haoran Jiang, Peng Lin, Xin Tang.

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
