## [Decision Letter · Decision Letter 0]

29 Sep 2025

Dear Dr. Wang,

Thank you for submitting your manuscript to PLOS ONE. After careful consideration, we feel that it has merit but does not fully meet PLOS ONE’s publication criteria as it currently stands. Therefore, we invite you to submit a revised version of the manuscript that addresses the points raised during the review process.

We look forward to receiving your revised manuscript.

Kind regards,

Yile Chen, Ph.D. in Architecture

Academic Editor

PLOS ONE

Journal Requirements:

3. Please note that PLOS One has specific guidelines on code sharing for submissions in which author-generated code underpins the findings in the manuscript. In these cases, we expect all author-generated code to be made available without restrictions upon publication of the work. Please review our guidelines at https://journals.plos.org/plosone/s/materials-and-software-sharing#loc-sharing-code and ensure that your code is shared in a way that follows best practice and facilitates reproducibility and reuse.

5. We note that Figure(s) 1, 6, 7, 8, 9 in your submission contain copyrighted images. All PLOS content is published under the Creative Commons Attribution License (CC BY 4.0), which means that the manuscript, images, and Supporting Information files will be freely available online, and any third party is permitted to access, download, copy, distribute, and use these materials in any way, even commercially, with proper attribution. For more information, see our copyright guidelines: http://journals.plos.org/plosone/s/licenses-and-copyright.

a. You may seek permission from the original copyright holder of Figure(s) 1, 6, 7, 8, 9 to publish the content specifically under the CC BY 4.0 license.

6. We note that Figure(s) 1, 6, 7, 8, 9 includes an image of a [patient / participant / in the study].

Reviewers' comments:

Reviewer's Responses to Questions

**Comments to the Author**

1. Is the manuscript technically sound, and do the data support the conclusions?

Reviewer #1: Partly

Reviewer #2: Partly

2. Has the statistical analysis been performed appropriately and rigorously?

Reviewer #1: No

Reviewer #2: Yes

3. Have the authors made all data underlying the findings in their manuscript fully available?

Reviewer #1: No

Reviewer #2: Yes

4. Is the manuscript presented in an intelligible fashion and written in standard English?

Reviewer #1: No

Reviewer #2: Yes

Reviewer #1: The Manuscript " Application of LHAT-YOLO Model in Engineering: Research on Intelligent Monitoring Algorithm for Helmets at Construction Sites" conducts a study on the helmets detection algorithm based on Yolo11. Through comparisons with different Yolo algorithms and experiments on different datasets, the advantages of the improved algorithm are verified but the advantages are not clearly. Generally speaking, this paper has a certain degree of academic value, but the innovation is insufficient. The paper needs to be significantly revised in the following aspects:

1. The title is too long and the first half of the title can be deleted. Generally, "study" should be used instead of "research" in the title.

2. The last paragraph of the introduction does not sufficiently summarize the innovation. The overall innovation of the manuscript is insufficient, and the manuscript does not reflect the issue of systematic real-time performance.

3. In Figure 5, the modification of YOLOV11 mainly lies in where it is located, but the representation in the figure is insufficient.

4. From the perspective of ablation experiments in Table 2, the improvement in algorithm performance is not obviously significant overall? And in the column of parameter quantity in Table 2, 6 should be an exponent.

5. In fact, there are many studies on safety helmet detection algorithms. This manuscript only compared with other Yolo algorithms in Table 3, and did not compare with other optimization algorithms based on Yolo. Therefore, it cannot fully reflect the innovation of the author's work.

6. The summary of the entire manuscript in the conclusion is not simple enough. The title is "Engineering Application ", but this point was not reflected at the end.

Reviewer #2: This manuscript proposes LHAT YOLO, a lightweight variant of YOLOv11 for detecting hardhat use on construction sites. The authors replace parts of the YOLOv11 backbone with GSConv and introduce a new Fast Convolutional Detection (FCD) head that includes “difference convolutions” to emphasize local gradients. On a self compiled dataset of 24,726 images labeled “helmet” and “person,” they report 11% lower GFLOPs and 9.5% fewer parameters than YOLOv11n, with precision 93.95%, recall 88.99%, mAP@50 94.92%, and mAP@50–95 65.28%. Qualitative examples suggest better robustness at night, under occlusion, and in clutter. However, this manuscript must be improved more for publication.

The manuscript can be enhanced by addressing the following comments:

1. Abstract is missing thorough review of applications of the most recent AI techniques and drones in safety helmet detection, for example, application of Transformer in M.Z. Shanti et al. Enhancing worker safety at heights: A deep learning model for detecting helmets and harness using DETR architecture, IEEE Access, 2025, pp151788-151802, and application of drones in M.Z. Shanti et al. Real-time monitoring of work-at-height safety hazards in construction sites using drones and deep learning, Journal of Safety Research, 2022, 83, pp364-370.

2. The FCD description is conceptual, but reproducibility requires exact architectural details: in/out channels and strides per branch, kernel sizes, the composition and ordering of CDC/HDC/VDC/ADC, normalization types, activation function placement, and how outputs are fused (Fig. 4a–b; pages 13–14). Please provide a layer by layer specification or an open source config file. Also, cite the original “Central Difference Convolution” work and any sources for the horizontal/vertical/angular variants referenced on lines 134–136.

3. The manuscript repeatedly refers to adding a “complex object detection layer” to YOLOv11 (lines 140–146). Please specify whether you added an extra detection scale (e.g., P2 with stride 4 or 8) and how it interfaces with the neck. Include final detection strides, feature map sizes for 640×640 input, and the number of heads. Sentence to amend: “Due to the complex and changing environment at the construction site. The study chose to add a complex object detection layer…” Proposed: “To better detect small and occluded objects, we add an additional detection scale at [specify level, e.g., P2, stride 8], fed by [neck block(s)], yielding [K] detection heads at strides [s1, s2, s3,(s4)].”

4. On lines 72–75, GFLOPs is described as “the number of floating point operations performed per second by the model.” That defines FLOPS (rate), not FLOPs (count). Please correct to “theoretical number of floating point operations per forward pass (at 640×640).” Sentence to amend: “GFLOPs refers to the number of floating point operations performed per second by the model.” Proposed: “GFLOPs denotes the number of floating point operations per forward pass at the specified input size; we report theoretical complexity estimated from the architecture.”

5. The paper lists OS/GPU/CPU and two hyperparameters but omits batch size, optimizer (SGD/AdamW), scheduler (cosine, one cycle), momentum/β, label smoothing, augmentations (Mosaic/MixUp/HSV/flip), EMA, NMS IoU and confidence thresholds, loss composition, number of epochs, and random seeds (lines 91–95). Please add these details and provide a public repository with training scripts and model weights.

6. Provide the training performance of the model, e.g., the plots of loss vs. epochs.

7. Provide tests for various environmental and image conditions, i.e., foggy, blurred, etc.

8. Figures and text mention modules such as C2PSA, PSABlock, C3k2, SPPF in Fig. 5 without definitions in the main text. Please include short descriptions and citations for each non standard component. Also, unify “heat map/hot map” to “heatmap” throughout, e.g., lines 184–186 and lines 203–204. Sentence to amend: “Similarly, the hot map results also show…” Proposed: “Similarly, the heatmap results show…”.

9. There are several grammatical errors in methodological sentences. Example: “A ablation experiment was conducted…” (line 154). Sentence to amend: “A ablation experiment…” Proposed: “An ablation experiment…”. Also, “Due to the complex and changing environment at the construction site. The study chose to add…” (lines 140–142) should be one sentence with a clear subject. These occur in sections where precision matters for readers trying to reimplement the approach.

10. Add the dataset size and the baseline(s) used for the 11%/9.5% reductions, and replace “However,” before the metric list with “Overall,” to avoid the implication of a trade off that the results do not show (Abstract lines 14–21).

11. Provide examples where LHAT YOLO fails, not just YOLOv11 failures, to guide future work.

12. Use “mAP@50” and “mAP@50–95” or “mAP50 / mAP50–95” consistently.

**Do you want your identity to be public for this peer review?** For information about this choice, including consent withdrawal, please see our Privacy Policy

Reviewer #1: No

Reviewer #2: No

---

## [Author Response · Author response to Decision Letter 1]

6 Dec 2025

Manuscript Number: PONE-D-25-44125R1

Response to Reviewers

Dear editors and reviewers,

Thank you for giving us the opportunity to submit a revised draft of the manuscript “Application of LHAT-YOLO Model in Engineering: Research on Intelligent Monitoring Algorithm for Helmets at Construction Sites” for publication in the Plos One. The authors would like to thank the editors and reviewers for the time and effort they invested in providing valuable comments on our manuscript. We are deeply grateful to the reviewers for their insights and valuable suggestions for improvement. The authors have incorporated the reviewers' suggestions. These changes have been highlighted in the manuscript. Please see the point-by-point responses to the reviewers' comments in the blue text section below for specific changes.

To Editors

It is a great honor to be recognized by the editors. The authors would like to thank the editors for their contributions in the process of revising the manuscript. Authors try to improve the quality of each part of the manuscript as much as possible. Once again, thank you for your review and support.

Author response: Thank you very much for your valuable comments. The authors have revised the manuscript according to the requirements of the editors and reviewers to ensure that the manuscript meets PLOS ONE's style requirements

Author response: Thank you very much for your valuable comments. This manuscript has no dispute over its edition.

3. Please note that PLOS One has specific guidelines on code sharing for submissions in which author-generated code underpins the findings in the manuscript. In these cases, we expect all author-generated code to be made available without restrictions upon publication of the work. Please review our guidelines at https://journals.plos.org/plosone/s/materials-and-software-sharing#loc-sharing-code and ensure that your code is shared in a way that follows best practice and facilitates reproducibility and reuse.

Author response: Thank you very much for your valuable comments. All code and dataset materials can be obtained from the corresponding author.

Please confirm at this time whether or not your submission contains all raw data required to replicate the results of your study. Authors must share the “minimal data set” for their submission. PLOS defines the minimal data set to consist of the data required to replicate all study findings reported in the article, as well as related metadata and methods (https://journals.plos.org/plosone/s/data-availability #loc-minimal-data-set-definition).

Author response: Thank you very much for your valuable comments. All code and dataset materials can be obtained from the corresponding author.

5. We note that Figure(s) 1, 6, 7, 8, 9 in your submission contain copyrighted images. All PLOS content is published under the Creative Commons Attribution License (CC BY 4.0), which means that the manuscript, images, and Supporting Information files will be freely available online, and any third party is permitted to access, download, copy, distribute, and use these materials in any way, even commercially, with proper attribution. For more information, see our copyright guidelines: http://journals.plos.org/plosone/s/licenses-and-copyright.

a. You may seek permission from the original copyright holder of Figure(s) 1, 6, 7, 8, 9 to publish the content specifically under the CC BY 4.0 license.

Author response: Thank you very much for your valuable comments. In fact, there are no copyright issues with the image data in the manuscript. The image data used was provided by the co-author of the manuscript, Haoran Jiang.

6. We note that Figure(s) 1, 6, 7, 8, 9 includes an image of a [patient / participant / in the study].

As per the PLOS ONE policy (http://journals.plos.org/plosone/s/submission- guidelines#loc-human-subjects-research) on papers that include identifying, or potentially identifying, information, the individual(s) or parent(s)/guardian(s) must be informed of the terms of the PLOS open-access (CC-BY) license and provide specific permission for publication of these details under the terms of this license. Please download the Consent Form for Publication in a PLOS Journal (http://journals.plos.org/plosone/s/file?id=8ce6/plos- consent-form-english.pdf). The signed consent form should not be submitted with the manuscript, but should be securely filed in the individual's case notes. Please amend the methods section and ethics statement of the manuscript to explicitly state that the patient/participant has provided consent for publication: “The individual in this manuscript has given written informed consent (as outlined in PLOS consent form) to publish these case details”.

Author response: Thank you very much for your valuable comments. In fact, the image data in the manuscript does not have any copyright disputes. The image data used was provided by the co-author of the manuscript, Haoran Jiang. These images were all collected by the company at construction sites, and no permission documentation is required.

Author response: Thank you very much for your valuable comments. The authors will seriously consider whether all the literature is of reference value to ensure the rigor of scientific research.

Reviewer #1

The Manuscript " Application of LHAT-YOLO Model in Engineering: Research on Intelligent Monitoring Algorithm for Helmets at Construction Sites" conducts a study on the helmets detection algorithm based on Yolo11. Through comparisons with different Yolo algorithms and experiments on different datasets, the advantages of the improved algorithm are verified but the advantages are not clearly. Generally speaking, this paper has a certain degree of academic value, but the innovation is insufficient. The paper needs to be significantly revised in the following aspects.

Author response: We are very honored that our manuscript has received your recognition. The manuscript indeed still has many shortcomings. The authors will revise the manuscript according to your requirements. Below are our responses to the questions you raised. We hope that the authors' revisions and responses will meet with your approval. Thank you once again for taking the time out of your busy schedule to review our manuscript.

1. The title is too long and the first half of the title can be deleted. Generally, "study" should be used instead of "research" in the title.

Author response: Thank you very much for your valuable suggestions. The authors have removed the first part of the manuscript title as requested. Additionally, the term “study” has been changed to “research” in both the title and the main text of the manuscript. This will further enhance the readability of the manuscript. We appreciate your recommendations once again. The revised content is highlighted in red font in the manuscript and accompanied by annotations.

2. The last paragraph of the introduction does not sufficiently summarize the innovation. The overall innovation of the manuscript is insufficient, and the manuscript does not reflect the issue of systematic real-time performance.

Author response: We sincerely appreciate your valuable suggestions. The authors have rephrased the final paragraph of the introduction to fully summarize the innovative contributions of the entire paper. Once again, thank you for your suggestions. The revised content is highlighted in red font in the manuscript and accompanied by annotations.

3. In Figure 5, the modification of YOLOV11 mainly lies in where it is located, but the representation in the figure is insufficient.

Author response: We sincerely appreciate your valuable comments. The authors have revised Figure 5 to highlight the improved aspects of the model. Additionally, they have provided a more detailed description of Figure 5 to emphasize these enhancements. Once again, thank you for your comments. This will further enhance the readability of the manuscript. The revised content is highlighted in red font in the manuscript and accompanied by annotations.

4. From the perspective of ablation experiments in Table 2, the improvement in algorithm performance is not obviously significant overall? And in the column of parameter quantity in Table 2, 6 should be an exponent.

Author response: Thank you very much for your valuable comments. The number 6 here indeed refers to an index, and the authors have already made the necessary revisions. Since the model underwent lightweight improvements primarily aimed at reducing training time and decreasing the number of parameters, the performance gains were not significant. Once again, we appreciate your suggestions. This will further enhance the readability of the manuscript. The revised content is highlighted in red font in the manuscript and accompanied by annotations.

5. In fact, there are many studies on safety helmet detection algorithms. This manuscript only compared with other Yolo algorithms in Table 3, and did not compare with other optimization algorithms based on Yolo. Therefore, it cannot fully reflect the innovation of the author's work.

Author response: Thank you very much for your valuable comments. The authors considered this issue prior to submitting the manuscript. However, since we did not utilize public datasets, we could not directly compare our results with those reported in other authors' papers. The dataset designed in the manuscript was specifically intended to train a model for detecting hard hats in complex construction environments. Applying this model to ordinary, simple datasets would likely lead to overfitting. Therefore, the authors did not compare their results with those from other papers. Nevertheless, your suggestion is highly constructive. We appreciate your suggestions once again.

6. The summary of the entire manuscript in the conclusion is not simple enough. The title is "Engineering Application ", but this point was not reflected at the end.

Author response: We sincerely appreciate your valuable comments. First, the authors have refined the content of the Conclusion section. Second, the authors have removed the first part of the manuscript title as requested to more accurately reflect the core focus of this study and emphasize the fundamental research nature of the manuscript. Once again, thank you for your suggestions. The revised content is highlighted in red font in the manuscript and accompanied by annotations.

Reviewer #2

This manuscript proposes LHAT YOLO, a lightweight variant of YOLOv11 for detecting hardhat use on construction sites. The authors replace parts of the YOLOv11 backbone with GSConv and introduce a new Fast Convolutional Detection (FCD) head that includes “difference convolutions” to emphasize local gradients. On a self compiled dataset of 24,726 images labeled “helmet” and “person,” they report 11% lower GFLOPs and 9.5% fewer parameters than YOLOv11n, with precision 93.95%, recall 88.99%, mAP@50 94.92%, and mAP@50–95 65.28%. Qualitative examples suggest better robustness at night, under occlusion, and in clutter. However, this manuscript must be improved more for publication.

Author response: First, we sincerely thank you for your contribution to reviewing this manuscript. The authors are deeply honored that you have taken the time to evaluate our work. We will make further improvements to the manuscript as per your suggestions. Once again, thank you for your patience and thorough review.

1. Abstract is missing thorough review of applications of the most recent AI techniques and drones in safety helmet detection, for example, application of Transformer in M.Z. Shanti et al. Enhancing worker safety at heights: A deep learning model for detecting helmets and harness using DETR architecture, IEEE Access, 2025, pp151788-151802, and application of drones in M.Z. Shanti et al. Real-time monitoring of work-at-height safety hazards in construction sites using drones and deep learning, Journal of Safety Research, 2022, 83, pp364-370.

Author response: Thank you very much for your valuable comments. The manuscript indeed lacks an in-depth review of the latest artificial intelligence technologies and the application of drones in the field of safety helmet inspection. The authors have revised the abstract and introd

---

## [Decision Letter · Decision Letter 1]

15 Dec 2025

LHAT-YOLO : Study on Intelligent Monitoring Algorithm for Helmets at Construction Sites

PONE-D-25-44125R1

Dear Dr. Wang,

We’re pleased to inform you that your manuscript has been judged scientifically suitable for publication and will be formally accepted for publication once it meets all outstanding technical requirements.

Kind regards,

Yile Chen, Ph.D. in Architecture

Academic Editor

PLOS One

Additional Editor Comments (optional):

Reviewers' comments:

Reviewer's Responses to Questions

**Comments to the Author**

Reviewer #1: All comments have been addressed

Reviewer #2: All comments have been addressed

2. Is the manuscript technically sound, and do the data support the conclusions?

Reviewer #1: Yes

Reviewer #2: Yes

3. Has the statistical analysis been performed appropriately and rigorously?

Reviewer #1: Yes

Reviewer #2: Yes

4. Have the authors made all data underlying the findings in their manuscript fully available?

Reviewer #1: Yes

Reviewer #2: Yes

5. Is the manuscript presented in an intelligible fashion and written in standard English?

Reviewer #1: Yes

Reviewer #2: Yes

Reviewer #1: The author has revised and improved the paper according to the suggested corrections, and it is recommended for acceptance.

Reviewer #2: Authors addressed all the comments properly. Therefore, I recommend the acceptance of this manuscript.

**Do you want your identity to be public for this peer review?** For information about this choice, including consent withdrawal, please see our Privacy Policy

Reviewer #1: No

Reviewer #2: No

---

## [Editor Report · Acceptance letter]

PONE-D-25-44125R1

PLOS One

Dear Dr. Wang,

I'm pleased to inform you that your manuscript has been deemed suitable for publication in PLOS One. Congratulations! Your manuscript is now being handed over to our production team.

Kind regards,

on behalf of

Dr. Yile Chen

Academic Editor

PLOS One